# Category-Specific Stress Mindsets: Beliefs about the Debilitating versus Enhancing Effects of Specific Types of Stressors among Young Adults

**DOI:** 10.3390/bs13090709

**Published:** 2023-08-26

**Authors:** Elijah R. Murphy, Diana J. Cox, Feven Fisseha, Kathleen C. Gunthert

**Affiliations:** 1Department of Psychology, American University, Washington, DC 20016, USA; dc7605a@american.edu (D.J.C.); ff6599a@american.edu (F.F.); gunthert@american.edu (K.C.G.); 2Department of Psychology, University of Houston, Houston, TX 77204, USA

**Keywords:** stress, stress mindset, stress appraisals, discrimination, academic stress

## Abstract

Recently, research has shown that stress mindsets, or the degree to which people believe that stress is enhancing versus debilitating, impact the ways they process and react to stress. However, young adults encounter various forms of stress, which might elicit different stress mindsets. This study investigated (1) how much young adults think about specific types of stressors as they complete stress mindset measures and (2) how stress mindsets vary across stressor types. Method: Participants (*n* = 182) completed measures of general and category-specific stress mindsets (academic, interpersonal, identity-based, illness, societal, financial) and rated how much they thought of each category when completing the general mindset measure. Results: Academic stress was the most salient, and identity-based discrimination was the least salient as participants completed the stress mindset measure. Academic stress was perceived as the most stress-enhancing, and illness stressors were rated as the least stress-enhancing. Cisgender women reported stronger stress-is-debilitating mindsets for interpersonal and illness/injury-related stressors as compared with cisgender men. Conclusion: Stress mindset ratings in research studies might be weighted toward certain types of stressors. Further, young adults’ mindsets differ across different stressor categories. This nuance has implications for how we conceptualize stress mindset in interventions and research.

## 1. Introduction

In American society, one receives many messages about how stress should be avoided. Certainly, stress can be a risk factor for a host of poor mental and physical health outcomes, such as depression, cardiovascular disease, and academic outcomes [1,2,3,4]. Despite the almost exclusive cultural focus on the downsides of stress, research suggests that stress can be both detrimental and enhancing. For example, stress can facilitate growth, learning, and social connection and push us to connect with meaning in life [5,6]. The growing literature on stress mindsets suggests that the degree to which people believe that stress is debilitating versus enhancing has important impacts on outcomes such as health, performance, and stress reactivity [7,8,9].

However, not all stressors are alike, and some will certainly be more helpful than others. Young adults in college might believe that academic stress could facilitate performance but also believe that financial stress is particularly detrimental. There is very little research, however, on how perceptions of the enhancing role of stress might vary across different types of stressors. The present study examined whether (1) stress mindset ratings of young adults are driven by specific types of stressors (Do specific stressor categories come to mind when people are asked about stress mindsets?) and (2) different stressor categories (e.g., academic, interpersonal, identity-based discrimination, financial, societal, and illness) elicit different stress mindsets.

### 1.1. Stress and Stress Appraisals

In the research literature on stress, one finds countless ways of conceptualizing human stress, from stimulus-based approaches that focus on external, objective stressors to response-based models that focus on stress as a physiological or psychological reaction to external or cognitive demands [10]. Perhaps the most widely embraced model of human stress is Lazarus and Folkman’s (1984) transactional model, which conceptualizes stress as the dynamic transaction between a person and the environment [11]. In this model, stress results from an imbalance between the perceived demands of the situation and the perceived resources for coping. Cognitive appraisals are, therefore, key in mediating the process from environmental stress to psychological and physiological outcomes. Threat appraisals tend to increase anxiety, avoidance motivation, and hypothalamus–pituitary–adrenal (HPA) axis activity, whereas challenge appraisals tend to be associated with more adaptive outcomes [12]. Appraisals, then, are perceptions of stressors that are driven by features of both the environment and the person. At the person level, for example, stress appraisals tend to be influenced by personality, such as neuroticism [13], individual resources, or mental health [14]. One individual difference variable that likely has important effects on specific stress appraisals and, in turn, stress outcomes is stress mindset. 

### 1.2. Stress Mindsets 

A stress mindset is one’s set of beliefs about the impacts that stress has on health and performance [7]. To varying degrees, an individual can believe that the experience of stress has enhancing/beneficial effects, or they can believe that the stress experience has debilitating/negative effects [7]. Although stress certainly has negative effects at times, stress has also been associated with several benefits, including social bonding, motivation, performance, connection to meaning, and psychological growth [15,16,17,18,19,20,21]. The notion that certain levels of stress can promote optimal performance dates all the way back to 1908, as the Yerkes–Dodson law suggested that an optimal level of arousal promotes optimal performance on behavioral tasks and over- or under-arousal reduces task performance [15]. In terms of motivation, academic and general stress have been associated with higher intrinsic motivation to accomplish tasks [18]. Outside of the classroom, stress and stress evaluation have also been found to positively predict subsequent performance in meaningful, stressful situations such as flying a plane [21], playing sports [17], or performing motor tasks such as conducting surgery [20]. In these cases, evaluating stress as challenging predicted better performances, while the evaluation that a stressful situation is threatening predicted worse performances [17,20,21]. 

Stress can also promote social bonding. Through biological processes primarily related to the hormone oxytocin, stress can elicit a tend and befriend response [19]. This stress-based surge in oxytocin can drive humans towards others by increasing pro-social behaviors [22]. Through mechanisms such as increasing trusting behaviors [23], increasing cooperation [24], and generosity [25], oxytocin positively impacts social relationships. Despite the common belief that stress has primarily negative impacts on health and well-being, the existing literature demonstrates that there are several adaptive impacts of stress. Importantly, people vary in the degree to which they believe that stress can be helpful. 

Stress mindset is a broad schema about whether stress tends to result in harm (such as decreased productivity) or benefits (such as increased motivation). This broad schema is a belief system that frames how a person takes in everyday stressors. As such, stress mindsets are conceptually distinct from specific stressor appraisals, which are perceptions of specific life stressors that are driven by both person variables (e.g., stress mindsets, personality) and the environment (e.g., features of the event itself) [7]. Overall, then, a stress mindset influences stress responses and is conceptually distinct from other stress variables, such as specific stressor appraisals, the severity of the stress experience, and coping methods [7]. 

Research has shown that stress mindsets are important in determining responses to threatening and challenging circumstances. For example, a stress-is-enhancing mindset was associated with more adaptive physiological responses to acute stressors [26]. Further, mental health outcomes are worse when stress perceptions are particularly negative, especially for those with stress-is-debilitating mindsets [8]. College students with greater stress-is-debilitating mindsets are more likely to experience an increase in depression following an increase in stress [8]. Positive stress beliefs have also been associated with lower somatic symptoms and increased academic performance, physical health, and psychological well-being [27]. Stress mindsets have been associated with school burnout, with one study showing that business students who viewed stress as more debilitating were more likely to experience burnout [28]. These findings suggest that, in general, there are benefits to a stress-is-enhancing mindset. Indeed, because of these findings, there are emerging intervention efforts to impact stress mindsets to increase student resiliency [29,30].

However, one’s stress mindset might change depending on the type of stressor one is facing. This nuance might be important for emerging stress mindset interventions. There is little work that focuses on the nuance of specific categories of stressors; do mindsets change as we consider different types of stressors? It is possible, for example, that some people believe that academic stressors can be helpful, but illness (for example) is not particularly beneficial. 

### 1.3. Digging Deeper into Mindset Beliefs—Are Some Stressors More Enhancing than Others?

Stress mindsets have important effects, but the existing literature often treats stress as one broad concept without a more nuanced lens to consider whether people have different mindsets for different types of stressors. It is important to study the way in which the typical features of different stressor types might systematically change stress mindsets. For example, stressors that tend to be more chronic, uncontrollable, unfair, or unjust might be less likely to foster a stress-is-enhancing mindset. In such cases, it might be harder to tap into the positive aspects of stress or identify positive outcomes for such stressors. Indeed, recent research suggests that the controllability and duration of the stressor source impact attitudes about the enhancing or debilitating role that stress plays [31]. 

An example of a stressor that often has the features of chronicity, uncontrollability, and unfairness would be identity-based discrimination. Perceived racial discrimination has been associated with decreased sense of mastery, increased psychological distress, and negative mental and physical health outcomes [32,33,34,35]. This is not to say that growth, motivation, and connection do not stem from overcoming identity-based discrimination; there are plenty of examples of such growth. However, just because growth can occur from the experienced stress of discrimination does not mean that it is not more challenging to grow from these types of stressors as compared with other stressors. Similarly, illness stressors can be quite chronic and uncontrollable, so the balance of negative to positive effects of illness-related stress might be more weighted toward the negative than other types of stressors. Thus, it is possible that stress mindsets about identity-based discrimination or illness/injury-related stress are more negative as compared with other types of stressors. 

Further, if different stressor categories elicit different stress mindsets, it would be important to ask the question: What categories do college students tend to think of as they respond to stress mindset questionnaires? It could be that when people are asked to think about how stress is harmful or beneficial, they tend to think about specific types of stressors, such as academic stressors, but less about other stressors, such as illness. Indeed, there could be systematic between-person differences in the types of stressors that come to mind when answering these questions. For example, it is possible that those who experience identity-based discrimination or illness/injury-related stress think about these forms of stress when answering the questions about stress outcomes more than those who tend not to experience discrimination or illness. This could lead to differences in global stress mindset scores that are driven by systemic differences in different categories of stressors. Given the increase in research on stress mindsets using global stress mindset measures, it would be helpful to understand if participants tend to think of certain types of stressors and not others as they respond to questions about their stress mindsets. 

This exploratory study examined (1) whether specific stressors tend to come to mind when young adults in college are asked about stress mindsets and (2) whether they view different stressor categories as more or less enhancing/debilitating. 

## 2. Methods 

### 2.1. Participants 

Inclusion criteria included being 18 years of age or older, fluent in English, and a current university student or recent alumni (in the past six months). This study’s sample included 182 undergraduate students and recent alumni (age range—18 to 26) at a private, mid-size, urban, Mid-Atlantic university; 42 individuals were identified as a man at birth, and 134 individuals were identified as a woman at birth. There were 117 White or Caucasian, 14 Black or African American, 16 Asian, and 17 multiracial participants. A total of 11 participants identified as another racial category, and 7 individuals did not answer. With respect to ethnicity, the sample included 21 Hispanic participants, 155 non-Hispanic participants, and 6 individuals who did not answer. Regarding gender identity, 41 participants identified as a cisgender man, 131 participants identified as a cisgender woman, 1 participant identified as a transgender man, 1 participant identified as genderqueer/gender non-conforming, 2 participants identified as non-binary, and 6 individuals did not answer. Regarding sexual identity, 3 participants identified as lesbian, 28 participants identified as bisexual, 11 participants identified as queer, 128 participants identified as heterosexual, 6 participants identified as an “other sexual identity”, and 6 individuals did not answer. A total of 29.1% of participants were freshmen, 22% of participants were sophomores, 12.6% of participants were juniors, 20.9% of participants were seniors, 6% were graduate students, and 7.7% were recent alumni.

### 2.2. Measures 

Stress Mindset. The Stress-Mindset Measure—General (SMM-G) [8] was used to assess general beliefs about stress. The measure is comprised of eight items measured on a 5-point scale ranging from 0-strongly disagree to 4-strongly agree. Half of the items measure a stress-is-enhancing mindset, such as “Experiencing stress facilitates my learning and growth”, and the other half measure a stress-is-debilitating mindset, such as “Experiencing stress depletes my health and vitality”. SMM-G scores are obtained by reverse scoring each of the four stress-is-debilitating mindset questions and then taking the mean of all eight items. Higher scores represent a stronger belief that stress is enhancing. The SMM-G has good internal consistency (Cronbach’s alpha of 0.80). The SMM-G only has minimal correlations with measures of stress, suggesting that the SMM-G is not redundant with the experience of stress, and the SMM-G is a significant predictor of health and life satisfaction above and beyond the effects of stress and coping strategies [7]. 

Stressor Salience. Stressor salience represents the degree to which participants thought of different categories of stressors as they responded to questions about stress mindsets. Upon completing the SMM-G, participants were asked to rate the degree to which six different stressor categories (academic, interpersonal, identity-based discrimination, illness, financial, and societal-level stressors) came to mind while completing the measure on a 5-point scale ranging from 1-not at all to 5-extremely. The excerpt was as follows: “We asked you a number of questions about how stress impacts you. We are interested in understanding what types of stressors people are thinking of when answering these questions. Please rate the degree to which the following stressors were on your mind as you answered questions about the effects of stress”. The participants were given stressor categories as follows: (1) academic stressors (e.g., exams, papers, due dates), (2) interpersonal stressors (e.g., difficult situations with other people), (3) identity-based discrimination, harassment, or microaggressions (e.g., systemic disadvantages associated with identity or interpersonal discrimination associated with identity), (4) societal-level stressors (e.g., politics, public policy, conflicts), (5) financial stressors (e.g., student loans, debt, making ends meet), and (6) illness or injury-related stressors (e.g., health issues, genetic illnesses). We chose broad stressor categories that we thought would encompass many of the stressors encountered by young adults in college. We drew these categories from other stress studies that have used broad categorizations. For example, Gunthert et al. (2002) used a daily stress approach that focused on stressor categories of interpersonal stress, academic stress, and illness/injury (as well as work stress and fatigue) [36]. Many studies document the importance of financial stress in college students [37], so we included it as a category. Finally, we added societal-level and identity-based discrimination as categories that are also likely quite salient to college students amid the COVID-19 pandemic. 

Specific Category Stress Mindsets. To investigate participants’ beliefs about individual stressor types, we made slight variations to the Stress-Mindset Measure (SMM-G) [8] and repeated the eight-item measure for each stressor category, asking participants to specifically imagine stressors of six types (academic, interpersonal, identity-based discrimination, illness, financial, and societal-level), and then answer the stress mindset questions again for that specific category. For all six specific category scales, Cronbach’s alphas were good, ranging from 0.84 to 0.89.

Additional measures were collected as a part of the larger study, including measures of identity, depression, resilience, perceived stress, personality, and social support.

### 2.3. Procedure 

Students were recruited through an advertisement posted on social media sites and were compensated with a $10 gift card (*n* = 89). In an effort to increase the sample size, participants were also recruited from introductory psychology courses and were compensated with a research credit (*n* = 93). The survey was emailed via Qualtrics and took approximately 20–30 min to complete. Enrollment for this study was conducted from approximately February 2020 to December 2020 amid the COVID-19 pandemic. Study participation was conducted virtually as students were taking courses online and off campus. This study was approved by the university’s institutional review board.

## 3. Results

### 3.1. Primary Analyses

**General Stress Mindsets.** The SMM-G is scored by calculating the mean of all eight items; the mean was 1.75 (*SD* = 0.71).

**Stress Mindset—What Stressors Drive Responses?** Repeated measures analysis of variance (ANOVA) was used to test what stress categories were most salient to participants when they were asked about stress mindsets, using the ratings of how much each category came to mind as they completed the stress mindset measure. There were significant differences across categories (*F* = 32.17, * p * < 0.001 partial *η*^2^ = 0.162). As shown in Table 1, academic stressors were thought of significantly more than all other stress types (interpersonal, identity-based discrimination, societal, financial, and illness/injury), with interpersonal stressors the next most salient. Illness and identity-based discrimination were thought of the least. See Table 1 for pairwise comparisons of salience across categories.

Stress Mindsets for Specific Stressor Categories. A repeated measures ANOVA was used to test whether there are stress mindset differences across the various stressor categories. There were significant differences across categories (*F* = 24.97, *p* < 0.001, partial *η*^2^ = 0.13). Academic and societal stressors were perceived as significantly more stress-enhancing than interpersonal, identity-based discrimination, financial, and illness/injury-related stressors. Illness/injury-related stressors were perceived as the least enhancing of the stressor types. See Table 2 for detailed pairwise comparisons.

### 3.2. Exploratory Analyses

Gender Differences. It is possible that individual differences in stressor salience and category ratings are related to personal identities. Unfortunately, our race and ethnicity categories were too small to allow for comparisons, but we were able to test for gender effects. Independent samples *t*-tests were conducted to evaluate potential differences across gender. Given the lack of sufficient sample size for some gender identities (specifically, transgender men; *n* = 1, genderqueer/gender non-conforming; *n* = 1), analyses were conducted with samples with sufficient sizes, specifically cisgender men; *n* = 41, cisgender women; *n* = 131. There were no significant differences between cisgender men and cisgender women in stressor salience; they reported thinking of similar stressor categories as they completed the broad stress mindset measure. Analyses revealed that there were significant differences between cisgender men and cisgender women in their ratings of interpersonal stressors (*p* < 0.05, mean difference= 0.26, CI = 0.00:0.52) and illness/injury stressors (*p* < 0.05, mean difference = 0.26, CI = 0.02:0.51). In both cases, cisgender women had more negative stress mindset ratings as compared with cisgender men. Detailed statistics on the differences between cisgender men and cisgender women can be found in Table 3.

## 4. Discussion 

This study examined the types of stressors that college students think of when asked to rate their stress mindset and whether there are differences in perceptions of how enhancing versus debilitating specific stressor categories tend to be. The present findings provide evidence that not all stressors have equal influence in driving stress mindset ratings, and different stressor categories do elicit different stress mindsets, consistent with recent research [31]. 

There were significant differences in the degree to which specific stress categories were thought of when students were asked about their stress mindsets. Academic stressors drove stress mindset ratings the most, whereas identity-based discrimination stressors were thought of the least. It is unsurprising that academic stress is particularly on one’s mind in a sample of college students, and research suggests that academic performance is a particularly relevant stressor for college students [37]. These findings could also be explained by the actual items in the stress mindset measure. Four out of the eight items on the stress mindset measure, such as “Experiencing stress enhances/inhibits my performance and productivity”, might arguably be more relevant for performance-based stressors such as work or academics. It will be important for researchers and educators to understand that when participants rate stress mindsets, their ratings might be particularly influenced by their beliefs about academic or performance-based stressors.

The second aim was to test whether stressor categories elicit different stress mindsets. Scores were not the same across the six different stressor types. Illness, financial, and identity-based discrimination stressors were perceived to be the least stress-enhancing, whereas academic and societal stressors were perceived to be the most stress-enhancing. Perhaps there are some similarities in the features of the lowest categories: identity-based, financial, and illness/injury-related stressors. Although this study does not include data on stressor appraisals, it is possible that there is less perceived controllability in identity-based discrimination, financial hardship, or illness-related stressors. Stressors that are perceived to be less controllable are likely to be deemed as more threatening, given that people might feel less confident of resources to meet the demands of stress, leading to a less stress-enhancing mindset [26]. The fact that illness stressors were rated as particularly detrimental might also be explained by historical effects. This study took place during the COVID-19 pandemic when illness was especially uncontrollable and rampant. It would be interesting to compare these ratings to those made in a time that is not characterized by a global pandemic. 

Exploratory analysis revealed that there were significant gender differences related to the perception of interpersonal stressors and illness/injury-related stressors. In both cases, cisgender women had less stress-is-enhancing mindsets than cisgender men. This is perhaps unsurprising, considering research showing that girls and women are particularly susceptible to the negative effects of interpersonal stressors [38]. Research also suggests that women are more likely to experience chronic physical health conditions, such as arthritis, headaches, and allergies [39]. In the future, it would be helpful to examine whether cisgender women’s ratings of the debilitating effects of illness/injury-related stress is related to systematically different types of illness and injury experiences.

### Limitations 

The study had a few limitations. Most importantly, there was a relatively low sample size for each specific racial category (besides White). Ideally, we would test whether there were race and ethnicity effects in stress category ratings. One might expect, for example, that identity-based discrimination is much more salient and potentially seen as more damaging among people who are systematically more likely to experience the impacts of discrimination. Further, there are other identities that systemically face identity-based discrimination (e.g., sexual and gender identities, religion, or disability). Future research will need to ensure that there is a sufficient sample size of marginalized identity groups to understand how the meaning of specific categories might change and differentially impact how young adults complete the broad stress mindset measure.

The present study asked about broad stressor categories, so it is not clear how these findings would relate to stress mindsets in the context of real-world specific stressors as people navigate their everyday lives. Additionally, although we assessed identity-based discrimination as a stressor category, it could also be helpful to assess a broader category of identity-based stress, which could include non-discrimination-type stressors such as acculturation and language barriers. This sample also consisted of college students attending a private Mid-Atlantic university, which limits the generalizability of the presented findings. Additional research will be needed to increase the generalizability of these findings to young adults nationwide and globally. Lastly, the present study included the general stress mindset measure and specific stress mindset measure for six different stressor categories. Given that each of these measures contains eight items, boredom or fatigue effects could have interfered with concentration as the study progressed for some participants.

## 5. Conclusions

There are exciting research developments on changing stress mindsets to facilitate stress resiliency. In the coming years, we will likely see many studies that seek to understand how stress mindsets can be shifted to impact mental health, resiliency, burnout, and retention. However, in any intervention and research efforts, it will be important to understand some of the nuances in young adults’ beliefs about stress. The present study provides evidence that there might be limits to treating stress as one broad concept when talking about stress mindsets with young adults. Different stressor categories can (1) differ in the degree to which they drive broad stress mindset ratings and (2) elicit different stress mindsets. Exploratory analyses provide additional evidence that stress mindset perceptions may differ across genders for specific categories of stressors. It will be important to understand these differences in stress mindsets and consider how systemic differences in stressor features and personal identities can impact these ratings. Future experimental and longitudinal research is needed to examine how a stress-is-enhancing mindset may be a protective factor against negative stress outcomes. Additional research is needed to appropriately examine stress mindset and stressor salience differences across gender, race/ethnicity, and sexual orientation.

## Figures and Tables

**Table 1 behavsci-13-00709-t001:** Means/SDs of stressor salience and mean differences between salience of stressor categories.

	Academic	Interpersonal	Identity-Based Discrimination	Societal	Financial	Illness
Academic	4.27 (0.81)	1.04 ***	2.17 ***	1.62 ***	1.16 ***	1.79 ***
Interpersonal		3.2 (1.22)	1.13 ***	0.58 ***	0.12	0.75 ***
Identity-based			2.09 (1.23)	−0.55 ***	−1.01 ***	−0.38 **
Societal				2.65 (1.13)	−0.46 ***	0.17
Financial					3.11 (1.4)	0.63 ***
Illness						2.46 (1.27)

Note: The diagonal represents the means and SDs of stressor salience ratings for specific categories. The rest of the table provides pairwise comparisons for ratings of stressor salience across categories; values representing mean differences are shown. ** *p* < 0.01; *** *p* < 0.001.

**Table 2 behavsci-13-00709-t002:** Means/SDs of stressor category ratings and mean differences between specific stress mindsets.

	Academic	Interpersonal	Identity-Based Discrimination	Societal	Financial	Illness
Academic	1.77 (*0.77*)	0.40 ***	0.55 ***	0.01	0.60 ***	0.90 ***
Interpersonal		1.37 (0.73)	0.15 *	−0.40 ***	0.20 ***	0.49 ***
Identity-based Discrimination			1.22 (0.77)	−0.54 ***	0.05	0.35 ***
Societal				1.76 (0.70)	0.59 ***	0.88 ***
Financial					1.17 (0.74)	0.29 ***
Illness						0.88 (0.71)

Note: The diagonal represents the means and SDs of stress mindset ratings for specific categories. The rest of the table provides pairwise comparisons for ratings of stress mindsets across categories; values represent mean differences shown. * *p* < 0.05; *** *p* < 0.001.

**Table 3 behavsci-13-00709-t003:** Means/SDs of stressor category ratings across gender.

	Cisgender Men (*n* = 41)	Cisgender Women (*n* = 131)
General	1.74 (0.80)	1.76 (0.68)
Academic	1.76 (0.78)	1.77 (0.77)
Interpersonal	1.57 (0.76)	1.31 (0.72) *
Identity-based Discrimination	1.31 (0.77)	1.22 (0.77)
Societal	1.73 (0.78)	1.78 (0.69)
Financial	1.26 (0.80)	1.15 (0.72)
Illness	1.09 (0.78)	0.82 (0.68) *

Note: These values represent the means and SDs of stress mindset ratings for general stress and specific categories. A * represents a significant mean difference. * *p* < 0.05.

## Data Availability

Data are available upon request. Please email the first author.

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
