# Peer review of "Category-Specific Stress Mindsets: Beliefs about the Debilitating versus Enhancing Effects of Specific Types of Stressors among Young Adults"

_behavsci, 2023, doi:10.3390/bs13090709_

Round 1

Reviewer 1 Report

This paper assessed how much young adults think about specific types of stressors as they complete stress mindset measures (academic most common, identity-based least common) and how stress mindsets vary across stressor types (stress-enhancing mindset scores highest with academic and lowest with illness). These are novel and interesting findings as the authors report that previous literature has often treated stress as a single, broad concept. Unfortunately, these results are overshadowed by the considerable amount of time looking for differences between racial/ethnic groups which the sample sizes do not adequately allow for.

General Concept Comments:

The primary research questions provided novel information, however, major issues exist with the manuscript. First, the authors devoted a considerable amount of time to attempting to assess differences between racial/ethnic identity groups. While listed in the limitations, the small sample sizes for non-white were small and hindered the ability to assess differences. The focus on these analyses takes away from the major finding of the article which could be concisely reported in a brief report. Second, why were differences in gender not assessed? These samples sizes provide an opportunity to do so.

Stressor Salience categories: How were these 6 categories chosen? Based upon what previous literature? How were they described to survey participants? Major issue in “identity-based” vs “identity-based discrimination” – see below.

Inconsistency in “identity-based” [stress] versus “identity-based discrimination” – these are two separate concepts and cannot be used interchangeably. It is unclear if participants were asked just about “identity-based” or “identity-based discrimination” in stressor salience.

Specific Category Stress Mindsets: It would be nice to see these surveys included in an Appendix. Again, were the 6 categorical words just given to participants or were examples of potential stressors given? Did you pilot test these surveys? Cronbach alpha?

Table 1: Include n’s for each category. Although when I did that, the total was 196 participants (not the 182 mentioned in the methods), meaning some participants were counted in multiple groups. This is concerning. The authors should clearly communicate the small sample sizes of non-white race/ethnicity groups and make sure these numbers add up to total participants. Why was only race/ethnicity mentioned and not gender?

Figure 1: Misleading figure. Standard deviation bars need to be added and not just include means. Sample sizes for each race/ethnicity group should also be provided. Paragraph in text needs to be edited as well (see below in Specific Comments).

Specific Comments:

·       Page 1, Line 30: Two “, and”s included in one sentence.

·       Page 2, Line 88: Change “associated decreased” to “associated with decreased”.

·       Page 2, Line 90: Two “and”s in same sentence.

·       Page 2, Line 93: “s” underlined.

·       Page 2, Line 87-Page 3, Line 98: XXXX

·       Page 3, Line 132: Authors do not mention the need to reverse score some SMM-G items.

·       Page 3, Line 138: Why were these specific 6 stressors chosen? Based upon what previous literature? What descriptions for each category was given to participants?

·       Page 3, Line 141: Numerical values not given that correspond to “not at all” vs “extremely”.

·       Page 5, Line 181: Groups included as People of color (POC) have not been defined and total n has not been given. The reader should be given all this information to make sense of results.

·       Page 5, Line 182: Concerning: “identity-based discrimination” used here – is that what participants were asked? As stated above “identity-based” stressors and “identity-based discrimination” are not equivalent and it is confusing why the authors are stating this in the results.

·       Page 5, Line 185-187: Actual means (standard deviations) are needed here, along with sample sizes somewhere in this paragraph for readers to understand the differences in sample sizes.

Check for typos. Potential to increase flow of article exists.

Author Response

Reviewer 1 Comments: 

This paper assessed how much young adults think about specific types of stressors as they complete stress mindset measures (academic most common, identity-based least common) and how stress mindsets vary across stressor types (stress-enhancing mindset scores highest with academic and lowest with illness). These are novel and interesting findings as the authors report that previous literature has often treated stress as a single, broad concept. Unfortunately, these results are overshadowed by the considerable amount of time looking for differences between racial/ethnic groups which the sample sizes do not adequately allow for. 

General Concept Comments: 

The primary research questions provided novel information, however, major issues exist with the manuscript. First, the authors devoted a considerable amount of time to attempting to assess differences between racial/ethnic identity groups. While listed in the limitations, the small sample sizes for non-white were small and hindered the ability to assess differences. The focus on these analyses takes away from the major finding of the article which could be concisely reported in a brief report. Second, why were differences in gender not assessed? These samples sizes provide an opportunity to do so. 

Thank you for this feedback. We realize that the sample size was not adequate to test effects of race, and so we have removed these analyses from the paper. We also added that race effects would be important for future research in the discussion section (345-348). We appreciate the suggestion to test for gender effects. We have added these analyses (277-294).  

Stressor Salience categories: How were these 6 categories chosen? Based upon what previous literature? How were they described to survey participants? Major issue in “identity-based” vs “identity-based discrimination” – see below. 

We have now added where the stressor categories were drawn from (219-225). They were taken from other studies that used broad stress categories, and we added societal-level and identity-based discrimination as categories that were likely part of the experience of many young adults. We agree that we needed to clarify the issue of identity-based vs identity-based discrimination (see below). 

Inconsistency in “identity-based” [stress] versus “identity-based discrimination” – these are two separate concepts and cannot be used interchangeably. It is unclear if participants were asked just about “identity-based” or “identity-based discrimination” in stressor salience. 

Thank you for catching this mistake. Indeed, identity-based stress and identity-based discrimination are not the same construct, though they overlap. We have now clarified that we did indeed ask participants about “identity-based discrimination, harassment, or microaggressions (ex: systemic disadvantages associated with identity or interpersonal discrimination associated with identity). Throughout the manuscript, we are now more specific about this category.  

Specific Category Stress Mindsets: It would be nice to see these surveys included in an Appendix. Again, were the 6 categorical words just given to participants or were examples of potential stressors given? Did you pilot test these surveys? Cronbach alpha? 

Unfortunately, the Stress Mindset Measure is copyrighted and so we are not able to put it in an Appendix. However, we give much more detail on the measure now in the Methods section. We did give participants examples of stressors for each category. We now give the full item prompt for each of the six categories in the Methods section (191-196). We did not pilot test this survey. We did not calculate Cronbach’s alpha for the saliency ratings because each rating (single item) was treated as it’s own variable. For the specific category ratings, there were eight items used for each category, like the general Stress Mindset Measure. We now report the range of alphas for those specific category ratings (231-232).   

Table 1: Include n’s for each category. Although when I did that, the total was 196 participants (not the 182 mentioned in the methods), meaning some participants were counted in multiple groups. This is concerning. The authors should clearly communicate the small sample sizes of non-white race/ethnicity groups and make sure these numbers add up to total participants. Why was only race/ethnicity mentioned and not gender? 

Thank you. We no longer report race differences and so the original Table 1 is no longer in the manuscript.  

Figure 1: Misleading figure. Standard deviation bars need to be added and not just include means. Sample sizes for each race/ethnicity group should also be provided. Paragraph in text needs to be edited as well (see below in Specific Comments). 

Thank you. We no longer report race differences and so Figure 1 was removed from the manuscript.  

Specific Comments: 

  •        Page 1, Line 30: Two “, and”s included in one sentence.

Thank you for this comment. This change has been made  

  •        Page 2, Line 88: Change “associated decreased” to “associated with decreased”.

Thank you for this comment. This change has been made  

  •        Page 2, Line 90: Two “and”s in same sentence.

Thank you for this comment. This change has been made  

  •        Page 2, Line 93: “s” underlined.

Thank you for this comment. This change has been made  

  •        Page 2, Line 87-Page 3, Line 98: XXXX –

Thank you for this comment. This change has been made  

  •        Page 3, Line 132: Authors do not mention the need to reverse score some SMM-G items.

Thank you for this comment. This change has been made to the methods section (194-196) 

  •        Page 3, Line 138: Why were these specific 6 stressors chosen? Based upon what previous literature? What descriptions for each category was given to participants?

Thank you for this comment. Additional literature has been added to support usage of these 6 stressor categories and information regarding the description given to participants has been included (219-225) 

  •        Page 3, Line 141: Numerical values not given that correspond to “not at all” vs “extremely”.

Thank you for this comment. Numerical values have been added (207) 

  •        Page 5, Line 181: Groups included as People of color (POC) have not been defined and total n has not been given. The reader should be given all this information to make sense of results.

Thank you for this comment. This section has been removed from the manuscript entirely  

  •        Page 5, Line 182: Concerning: “identity-based discrimination” used here – is that what participants were asked? As stated above “identity-based” stressors and “identity-based discrimination” are not equivalent and it is confusing why the authors are stating this in the results.

Thank you for this important comment. This change has been made, the participants were asked about identity-based discrimination specifically so we have changed the wording throughout the manuscript .·     

  Page 5, Line 185-187: Actual means (standard deviations) are needed here, along with sample sizes somewhere in this paragraph for readers to understand the differences in sample sizes. 

Thank you for this comment. This section was removed from the manuscript and the portion that replaced this section includes sample sizes, means and standard deviations. (287-293) 

Reviewer 2 Report

Thank you for the opportunity to review your manuscript. I found the topic to be interesting and timely. Stress for young adults continues to be a need in research, especially following COVID-19 with stress continuing to be something impacting young adults. Your literature review provided a new focus on stress mindsets and specific stressors that may impact stress mindset. The purpose of the article is clearly stated on page 3.

Methods

Please include when the data was collected in your methods section to let readers know that it took place during COVID-19 and when it took place. Were the participants taking courses virtually instead of in person at the time of data collection? 

Participants and Setting

Please provide more information about the setting. I would include that it is a private institution in the setting section. What is the approximate population of the university? Where is located, urban, rural, etc.? Was your sample size representative of the student population of the university? I would also encourage you to include more information about the participants, such as age, grade level, first generation college student, etc. This information is important, because students at 18 and students in their mid-late 20s often have different access to resource, such as mental health and finances. Did you run any of your analyses for anything besides race?

Measures

I liked how you explained how many questions were included in the first measure, Stress Mindset. I would encourage you to include how many questions were asked for each question. The number of questions can often lead to survey fatigue, so this is good to include, especially if the questions are asked in the same order. This may be something to explore in the discussion section or limitations. 

I liked the use of tables to accompany the results descriptions.

Figure 1 appeared blurry. I would recommend reviewing the figure to see if you can make the text clearer.

Limitations

You explain that there was a low sample size of specific racial categories besides white, yet in your results you identify that people of color perceive identity-based description more than other groups. How did you account for the difference in sample size when running the ANOVAs?

I did notice some English errors. I would recommend reading through to identify grammatical errors that could be revised during the revision stage. 

Author Response

Reviewer 2 Comments: 

Thank you for the opportunity to review your manuscript. I found the topic to be interesting and timely. Stress for young adults continues to be a need in research, especially following COVID-19 with stress continuing to be something impacting young adults. Your literature review provided a new focus on stress mindsets and specific stressors that may impact stress mindset. The purpose of the article is clearly stated on page 3. 

Methods 

Please include when the data was collected in your methods section to let readers know that it took place during COVID-19 and when it took place. Were the participants taking courses virtually instead of in person at the time of data collection?  

This is a good point, thank you. We’ve now added the dates of data collection in the procedures section (239-241). The students were taking courses virtually at the time. We added this to the methods section (240-243).  

Participants and Setting 

Please provide more information about the setting. I would include that it is a private institution in the setting section. What is the approximate population of the university? Where is located, urban, rural, etc.? Was your sample size representative of the student population of the university? I would also encourage you to include more information about the participants, such as age, grade level, first generation college student, etc. This information is important, because students at 18 and students in their mid-late 20s often have different access to resource, such as mental health and finances. Did you run any of your analyses for anything besides race? 

We have now included more information about the population in the methods section (170-187). We have added exploratory analyses testing gender effects (270-294).  

Measures 

I liked how you explained how many questions were included in the first measure, Stress Mindset. I would encourage you to include how many questions were asked for each question. The number of questions can often lead to survey fatigue, so this is good to include, especially if the questions are asked in the same order. This may be something to explore in the discussion section or limitations.  

Thank you. We have now added the specific questions in the methods section (207-218). Fortunately, this survey tended to take only about 25 minutes per person, and so it was not excessively long or time consuming. For the specific category mindset questionnaires, we asked eight questions per category, for a total of 48 questions for the main outcome variable, now noted in the manuscript. We have noted in the limitations section that completing the mindset items repeatedly for each category could conceivably lead to fatigue or boredom (356-359).   

I liked the use of tables to accompany the results descriptions. 

Figure 1 appeared blurry. I would recommend reviewing the figure to see if you can make the text clearer. 

Because we cut the race analyses, Figure 1 is no longer in the manuscript. 

Limitations 

You explain that there was a low sample size of specific racial categories besides white, yet in your results you identify that people of color perceive identity-based description more than other groups. How did you account for the difference in sample size when running the ANOVAs? 

Thank you. We have cut the race-based analyses from our manuscript.  

Reviewer 3 Report

The manuscript entitled “Category-specific stress mindsets: Beliefs about the debilitating versus enhancing effects of specific types of stressors among young adults” is a valuable and engaging contribution to the literature on mental health. The study described in the manuscript relies on young adults’ self-reports of stress and its diverse sources. Its results are likely to be of interest to a broad spectrum of clinicians and scholars. In my modest opinion, the manuscript merits publication after a few concerns are addressed.  The following are some of these concerns.

 The abstract needs to be rewritten to ensure that its content is clear and appealing to a broad array of readers. The text involving the authors’ conclusions is not clear at all. The authors specifically state that “[s]tress mindset ratings might be weighted toward certain types of stressors. Further, young adults’ mindsets differ across different stressor categories. This nuance has implications for stress mindset discussion in intervention and research”. Two sentences seem to be repeating some of the findings. It is unclear what is “stress mindset discussion in intervention and research”. The authors need to be clear about the implications and/or applications of their findings.

 The introduction needs to be expanded to cover more comprehensively the physiological and psychological aspects of stress, precursors, desirable and undesirable consequences, and known clinical interventions. Most importantly, the rationale for each of the dependent measures selected needs to be put forth. The authors state that they measured stress mindset, stressor salience, and specific category stress mindsets. Yet, they fail to offer a clear rationale for their study. What is the hypothesis associated with each dependent measure? Why did the authors focus on young adults?

In the method section, the procedure used to recruit the participants and the information given to them regarding the study must be clarified. Readers are told that “[s]tudents were recruited through an advertisement posted on social media sites and were compensated with a $10 gift card (n = 89). We also sampled introductory psychology students who were compensated with a research credit (n = 93). What was the rationale for gathering two samples? Is there a reason to assume that these two groups of students are equivalent?

The demographic characteristics collected by the authors leave readers to ask whether particular cultural dispositions may account for the study’s findings. A review of the cross-cultural literature may offer a window into this issue as well as contribute to a more comprehensive discussion of the study’s findings.

Is the sample size adequate for the statistical analyses carried out? Given the small sample size, did the author consider bootstrapping?

Minor editing of English language is required

Author Response

Reviewer 3 Comments: 

The manuscript entitled “Category-specific stress mindsets: Beliefs about the debilitating versus enhancing effects of specific types of stressors among young adults” is a valuable and engaging contribution to the literature on mental health. The study described in the manuscript relies on young adults’ self-reports of stress and its diverse sources. Its results are likely to be of interest to a broad spectrum of clinicians and scholars. In my modest opinion, the manuscript merits publication after a few concerns are addressed.  The following are some of these concerns. 

 The abstract needs to be rewritten to ensure that its content is clear and appealing to a broad array of readers. The text involving the authors’ conclusions is not clear at all. The authors specifically state that “[s]tress mindset ratings might be weighted toward certain types of stressors. Further, young adults’ mindsets differ across different stressor categories. This nuance has implications for stress mindset discussion in intervention and research”. Two sentences seem to be repeating some of the findings. It is unclear what is “stress mindset discussion in intervention and research”. The authors need to be clear about the implications and/or applications of their findings. 

Thank you, our abstract has been reworded to increase clarity (9-13). 

 The introduction needs to be expanded to cover more comprehensively the physiological and psychological aspects of stress, precursors, desirable and undesirable consequences, and known clinical interventions. Most importantly, the rationale for each of the dependent measures selected needs to be put forth. The authors state that they measured stress mindset, stressor salience, and specific category stress mindsets. Yet, they fail to offer a clear rationale for their study. What is the hypothesis associated with each dependent measure? Why did the authors focus on young adults? 

Thank you. We have expanded the introduction section to include a broader set-up of stress theory, the psychological aspects of stress, and stress outcomes (47-85). We further elaborated on stress mindset theory and some of the potential benefits of positive stress mindsets in order to strengthen our rational for our study. The stress mindset measure was originally validated in college students, and stress mindset interventions have targeted young adults and college students (86-101).  

In the method section, the procedure used to recruit the participants and the information given to them regarding the study must be clarified. Readers are told that “[s]tudents were recruited through an advertisement posted on social media sites and were compensated with a $10 gift card (n = 89). We also sampled introductory psychology students who were compensated with a research credit (n = 93). What was the rationale for gathering two samples? Is there a reason to assume that these two groups of students are equivalent? 

Thank you. This study was conducted early in the COVID-19 pandemic. This was a time when study recruitment was difficult, and so we recruited participants from multiple sources in order to boost our sample size. The 93 participants compensated with research credit were taking introductory psychology and they are asked to complete research credits as part of their introduction to psychology. To boost recruitment, we also offered gift card compensation to students who were not enrolled in introductory psychology (237-243). Because they are all students or very recent graduates from the same University, there is no strong reason to suspect that the groups are not similar and should not be combined into one sample.    

The demographic characteristics collected by the authors leave readers to ask whether particular cultural dispositions may account for the study’s findings. A review of the cross-cultural literature may offer a window into this issue as well as contribute to a more comprehensive discussion of the study’s findings. 

Is the sample size adequate for the statistical analyses carried out? Given the small sample size, did the author consider bootstrapping? 

Thank you. We have cut the race-based analyses from our manuscript. We agree that the sample size was not adequate for the statistical analyses performed.